# Stabilization of GTSE1 by cyclin D1–CDK4/6-mediated phosphorylation promotes cell proliferation with implications for cancer prognosis

Nelson García-Vázquez[1], Tania J González-Robles[1,2,3], Ethan Lane[1], Daria Spasskaya[1], Qingyue Zhang[1], Marc A Kerzhnerman[1], YeonTae Jeong[1†], Marta Collu[1], Daniele Simoneschi[1], Kelly V Ruggles[2], Gergely Róna[1,3,4], Sharon Kaisari[1,3]*, Michele Pagano[1,3]*

[1]Department of Biochemistry and Molecular Pharmacology, New York University Grossman School of Medicine, New York, United States; [2]Department of Medicine, New York University Grossman School of Medicine, New York, United States; [3]Howard Hughes Medical Institute, New York University Grossman School of Medicine, New York, United States; [4]Institute of Molecular Life Sciences, HUN-REN Research Centre for Natural Sciences, Budapest, Hungary

**\*For correspondence:**
sharon.kaisari@nyulangone.org (SK);
michele.pagano@nyulangone.org (MP)

**Present address:** †Arvinas Inc, New Haven, United States

## eLife Assessment

In this article, García-Vázquez et al. report **valuable** findings demonstrating that G2 and S phases expressed protein 1 (GTSE1), is a previously unappreciated non-pocket substrate of the cyclin D/ cyclin-dependent kinase (CDK) 4/6 axis. The authors provide **convincing** evidence showing that this mechanism is triggered in pathological states in which cyclin D levels are elevated (e.g., cancer). Overall, this study holds a promise to improve understanding of the mechanisms underpinning cell cycle progression including its dysregulation in neoplasia and may thus be of broad interest to researchers belonging to diverse biomedical disciplines ranging from cancer research to cell biology.

**Abstract** In healthy cells, cyclin D1 is expressed during the G1 phase of the cell cycle, where it activates CDK4 and CDK6. Its dysregulation is a well-established oncogenic driver in numerous human cancers. The cancer-related function of cyclin D1 has been primarily studied by focusing on the phosphorylation of the retinoblastoma (RB) gene product. Here, using an integrative approach combining bioinformatic analyses and biochemical experiments, we show that GTSE1 (G-Two and S phases expressed protein 1), a protein positively regulating cell cycle progression, is a previously unrecognized substrate of cyclin D1–CDK4/6 in tumor cells overexpressing cyclin D1 during G1 and subsequent phases. The phosphorylation of GTSE1 mediated by cyclin D1–CDK4/6 inhibits GTSE1 degradation, leading to high levels of GTSE1 across all cell cycle phases. Functionally, the phosphorylation of GTSE1 promotes cellular proliferation and is associated with poor prognosis within a pan-cancer cohort. Our findings provide insights into cyclin D1's role in cell cycle control and oncogenesis beyond RB phosphorylation.

## Introduction

D-type cyclins (cyclin D1, cyclin D2, and cyclin D3) are activators of the cyclin-dependent kinases CDK4 and CDK6 and represent major oncogenic drivers among members of the cyclin superfamily (*Musgrove et al., 2011*; *Fassl et al., 2022*; *Álvarez-Fernández and Malumbres, 2020*; *Behan et al., 2019*). The *CCND1* gene `--encoding cyclin D1--` shows some of the highest frequency of amplification and overexpression among cancer genes across a broad spectrum of human tumors (*Beroukhim et al., 2010*; *Curtin et al., 2005*; *Bignell et al., 2010*; *Parsons et al., 2008*; *Jones et al., 2008*). Moreover, mutations in *CCND1*, which prevent the degradation of cyclin D1 by AMBRA1, a substrate receptor (SR) of a CUL4-RING ubiquitin ligase (CRL4) complex that targets all three D-type cyclins for proteasome-mediated degradation, have been reported in a variety of tumor types (*Simoneschi et al., 2021*; *Chaikovsky et al., 2021*; *Martínez-Jiménez et al., 2020*). Deregulation of cellular proliferation, often mediated by uncontrolled CDKs' activation lies at the heart of cancer as a pathological process (*Kastan and Bartek, 2004*).

The biological roles of D-type cyclins have been examined almost exclusively through the lens of E2F transcription regulation upon phosphorylation of the three pocket proteins, RB, p107, and p130. D-type cyclin-mediated phosphorylation of RB results in the inactivation of its tumor suppressive effect by releasing E2F from the RB's inhibitory effect on gene transcription. Although the regulation of RB by phosphorylation is well understood in both physiology and human tumorigenesis, the cancer-related role of phosphorylation of other substrates is a subject that has remained understudied. In this study, we characterize GTSE1 (G-Two and S phase expressed protein 1) as a previously unidentified substrate of cyclin D1–CDK4/6 in cells overexpressing cyclin D1 in G1 and beyond. GTSE1 is a cell cycle-related protein expressed specifically during the S and G2 phases of the cell cycle (*Liu et al., 2010*). It interacts with the tumor suppressor p53, and it induces its MDM2-mediated degradation in G2 and during the recovery from DNA damage, promoting cell proliferation (*Liu et al., 2010*; *Bublik et al., 2010*; *Monte et al., 2003*; *Monte et al., 2004*; *Collavin et al., 2000*; *Utrera et al., 1998*). Moreover, GTSE1 associates with growing microtubules, promoting cell migration (*Bendre et al., 2016*; *Scolz et al., 2012*). In prometaphase, GTSE1 becomes highly phosphorylated by CDK1–cyclin B1, resulting in its recruitment to the inner spindle (*Singh et al., 2021*). After anaphase, its dephosphorylation is followed by marked reduction in its abundance in G1 (*Collavin et al., 2000*; *Monte et al., 2000*). Our findings demonstrate that the cyclin D1–CDK4/6-mediated phosphorylation of GTSE1 leads to its increased stability in G1 phase, an event that significantly impacts cell proliferation and cancer prognosis.

## Results and discussion

AMBRA1 is an SR of a CUL4-RING ubiquitin ligase (CRL4) complex that targets all three D-type cyclins for proteasome-mediated degradation (*Simoneschi et al., 2021*; *Chaikovsky et al., 2021*). To unveil new substrates of the cyclin D1–CDK4 complex, we performed a comprehensive multi-analysis of published data (*Figure 1—figure supplement 1A*). First, we incorporated recent mass spectrometry data that compared the whole proteome of AMBRA1 knockout (KO) clones to parental U2OS cells (*Chaikovsky et al., 2021*), aiming to pinpoint the top 30 proteins whose levels were elevated in the absence of AMBRA1, the SR of the ubiquitin ligase targeting D-type cyclins (*Figure 1A*). Subsequently, we asked which of these 30 proteins contain a canonical CDK phosphorylation consensus motif [S/T*] PX[K/R], relying on the PhosphoSitePlus database (*Hornbeck et al., 2015*; *Figure 1B*). Finally, this dataset was integrated with findings from a proteomic screen assessing protein abundance fluctuations in the presence or absence of the CDK4/6 inhibitor, Palbociclib (*Figure 1C*; *Chaikovsky et al., 2021*). In summary, our analysis aimed at identifying proteins whose upregulation in AMBRA1 KO cells was counteracted by Palbociclib treatment, thereby filtering for proteins whose augmented abundance is attributed to CDK4/6-mediated phosphorylation events when cyclin D1 is stabilized. GTSE1, a cell cycle-regulated protein expressed mainly during the G2 and S phases of the cell cycle (*Liu et al., 2010*), emerged as the top hit in these orthogonal analyses. In mitosis GTSE1 is phosphorylated by CDK1–cyclin B1 (*Singh et al., 2021*), but its dephosphorylation at the end of mitosis is followed by marked reduction in its abundance, which remains low during the next G1 (*Collavin et al., 2000*; *Monte et al., 2000*).

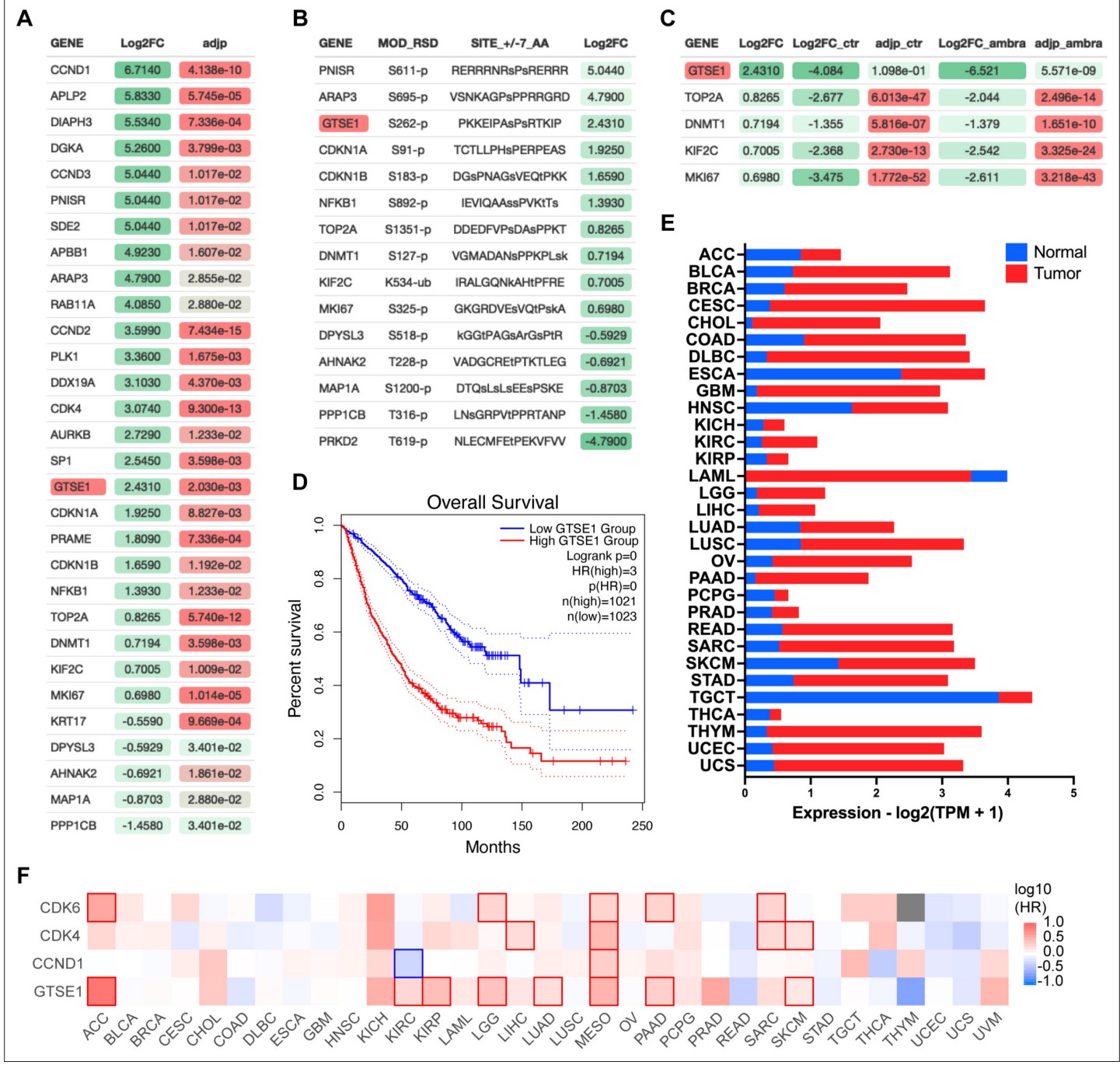

**Figure 1.** Identification of GTSE1 as a cyclin D1–CDK4 substrate with prognostic significance in cancer. (**A**) Differential proteomic profiling of AMBRA1 knockout (KO) U2OS cells compared to parental U2OS cells (**Chaikovsky et al., 2021**). Proteins with adjusted p-value <0.05 were ranked by the log2 fold change (Log2FC) to quantify expression changes between AMBRA1 KO and parental cells. The top 30 upregulated proteins in this analysis are presented. Statistical significance was assessed by false discovery rate (FDR). (**B**) Subset of proteins from (**A**) containing a canonical CDK phosphorylation consensus motif [S/T*]PX[K/R]. Annotated list of phosphorylated proteins was generated using PhosphoSitePlus database (**Hornbeck et al., 2015**). (**C**) Subset of proteins from (**B**) whose abundance was reverted to basal levels following treatment of AMBRA1 KO U2OS cells with Palbociclib (**Chaikovsky et al., 2021**) (Log2FC_ambra) with adjusted p-values (adjp_ambra) <0.001 were ranked by log2 fold change (Log2FC) from the original shotgun proteomic analysis (**A**) to integrate expression changes between AMBRA1 KO and parental cells under Palbociclib-treated versus untreated condition. (**D**) Kaplan–Meier curves representing the overall survival analysis based on the 50% upper versus lower expression levels of GTSE1. Survival analysis was conducted across various cancer cohorts, including ACC, KIRP, KIRC, LGG, LUAD, MESO, PAAD, and SKCM (see also **Figure 1—figure supplement 1**). The hazard ratio (HR) was calculated to estimate the relative risk, and significance was assessed using the log-rank test with a threshold of p < 0.05. Curves were generated using the GEPIA2 platform (**Tang et al., 2019**). (**E**) Transcriptomic analysis of GTSE1 in various

*Figure 1 continued on next page*

Figure 1 continued

cancer types, contrasting tumor (red) and normal tissue (blue) expression levels, using data derived from The Cancer Genome Atlas (TCGA) (*The Cancer Genome Atlas Research Network et al., 2013*). Statistical significance assessed by FDR, p < 0.05. (**F**) Survival map of various cancer types according to the indicated gene expression levels (50% upper and lower expression). Color intensity indicates the log of the HR, with red and blue representing poorer and better survival, respectively. Statistically significant differences in survival are denoted by highlighted contour squares, based on the log-rank test. Generated with GEPIA2 platform (*Tang et al., 2019*), using TCGA data.

The online version of this article includes the following figure supplement(s) for figure 1:

**Figure supplement 1.** Identification of GTSE1 as a D-type cyclin substrate and Its prognostic significance in cancer.

Recent pan-cancer analysis revealed that the expression of GTSE1 positively correlates with tumor mutational burden and microsatellite instability in most cancer types (*Tan et al., 2023*). Specifically, high expression of GTSE1 was found to promote the proliferation and invasion of breast cancer cells (*Lin et al., 2019*), and was associated with poor clinical prognosis in clear cell renal cell carcinoma (*Lei et al., 2023*). We explored further the potential influence of GTSE1 on cancer prognosis and found that within multiple cancer cohorts, its elevated expression levels were correlated with a statistically significant poorer prognosis compared to lower expression levels (*Figure 1D*, *Figure 1—figure supplement 1B*). This is noteworthy, since, except for acute myeloid leukemia (LAML), GTSE1 expression was higher than that in normal tissue counterparts in all tumor types analyzed (*Figure 1E*). We also assessed the impact of GTSE1 on survival across various cancer types (*Figure 1F*, *Figure 1—figure supplement 1B*). For context, we compared the survival patterns associated with cyclin D1, CDK4, and CDK6. Similar to the overexpression of other cell cycle-regulated genes (*Whitfield et al., 2006*), high levels of GTSE1 were found to be prognostically unfavorable across multiple cancer types, presenting a survival pattern similar to that of the cyclin D1–CDK4/6 gene cluster. In *Figure 1F*, those cancers in which differences in survival are statistically significant were denoted by contour squares.

Next, we sought to experimentally validate GTSE1 as a putative phosphorylation target of cyclin D1–CDK4. As a first step, we employed transient transfection to introduce in HEK293T various Flag-tagged constructs in the presence or absence of cyclin D1 and CDK4. GTSE1 showed an upper shift in a phos-tag gel when co-expressed with cyclin D1–CDK4, similar to the known substrates p107 and p130 (*Figure 2A*). In fact, a slight delay in GTSE1 migration is appreciable even in a regular SDS–PAGE (see first two lanes in *Figure 2A*). The upper shift was not observed in ZC3HAV1, a protein used as negative control. Next, we performed an in vitro phosphorylation assay to directly assess the specificity of GTSE1 phosphorylation by cyclin D1–CDK4. Using purified, recombinant proteins, we subjected GTSE1 to a phosphorylation reaction with cyclin D1–CDK4 and analyzed the products using phos-tag gels. The results confirmed that GTSE1 underwent phosphorylation by cyclin D1–CDK4, but not by ERK1, another Pro-directed kinase (*Figure 2B*). Moreover, the phosphorylation was abolished in the presence of Palbociclib, similar to the pattern observed with RB (*Figure 2B, C, H*). Then, we asked whether GTSE1 physically interacts with the kinase complex. Overexpression of Flag-tagged cyclin D1 and CDK4 in HEK293T followed by a Flag pull-down demonstrated binding to endogenous GTSE1, similar to the canonical substrate RB (*Figure 2D*).

Following these validation steps, we aimed at pinpointing the serine residue(s) modified by cyclin D1–CDK4. First, we generated a GTSE1 mutant in which Ser262 (see *Figure 1B*) was mutated to Ala. Since GTSE1 has additional, conserved serine residues followed by prolines, which could be potential CDK phosphorylation sites (*Figure 2E*, *Figure 2—figure supplement 1A, B*), we also generated another sixteen Ser-to-Ala GTSE1 mutants and expressed them in HEK293T cell in the presence or absence of cyclin D1–CDK4 (*Figure 3—figure supplement 1A*). When compared to wild-type GTSE1, two mutants displayed changes in their migrations on phos-tag gels (*Figure 3—figure supplement 1A*), suggesting that Ser residues at positions 91 and 724 are sites potentially phosphorylated by cyclin D1–CDK4. PhosphoSitePlus (*Hornbeck et al., 2015*) reports several high-throughput studies that, in addition to Ser262, identified phosphorylation also at Ser91 and Ser724, supporting our results. To corroborate the loss of cyclin D1–CDK4-dependent phosphorylation at these sites, we constructed a triple mutant (S91A/S262A/S724A). This triple mutant displayed loss of slower-migrating bands relative to wild-type GTSE1, suggesting diminished phosphorylation (*Figure 2F*). Nevertheless, a residual slow-migrating band persisted, prompting further mutations of the triple GTSE1 mutant in two additional GTSE1 sites (individually), which do not have a CDK-phosphorylation consensus, but were identified in several proteomics studies (*Hornbeck et al., 2015*; *Kaulich et al., 2021*). From these two

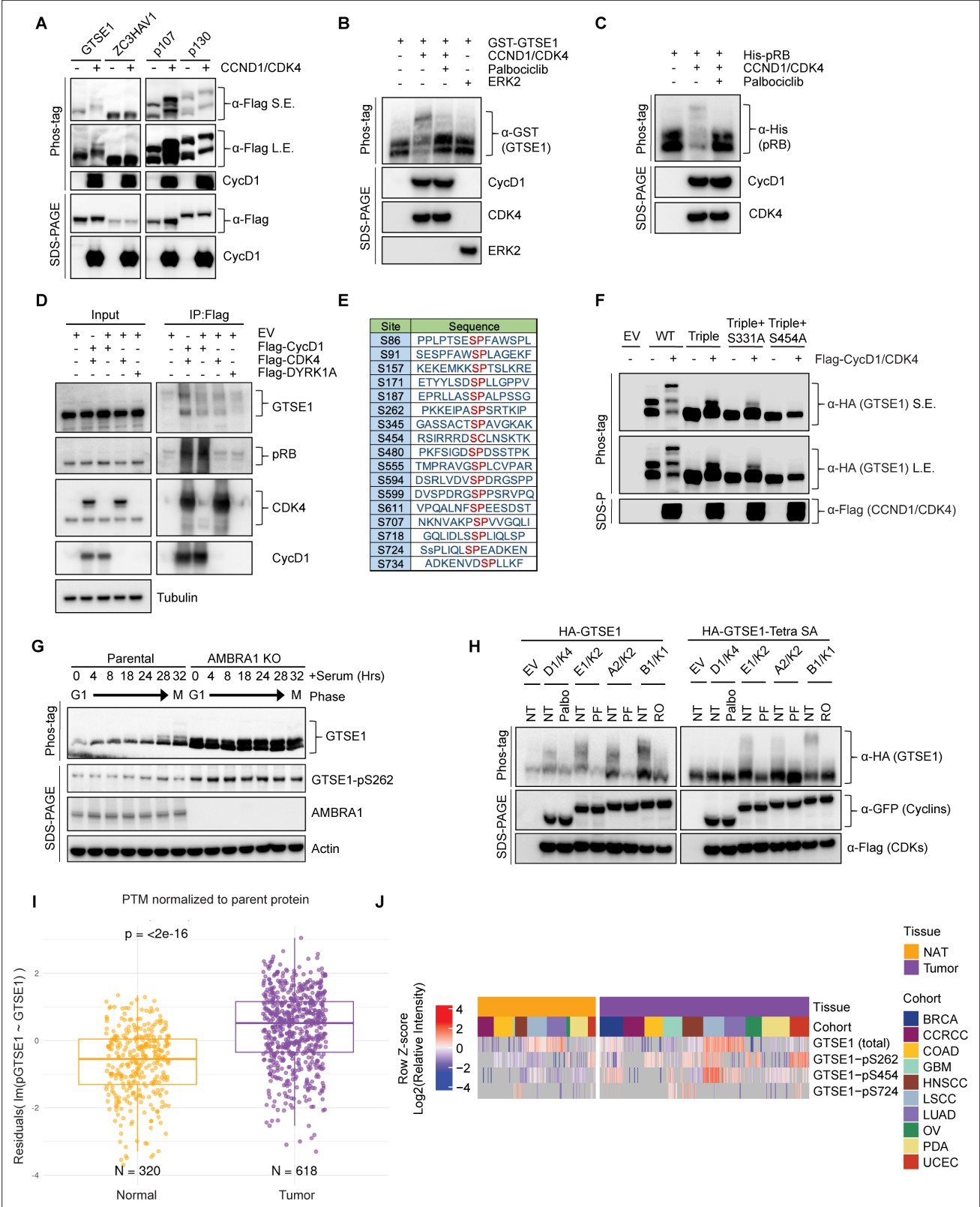

**Figure 2.** Cyclin D1–CDK4 promotes GTSE1 phosphorylation on four serine residues. (**A**) Immunoblot analysis following transient transfection of the indicated proteins in HEK293T cells detailing their phosphorylation status via differential mobility in phos-tag gels in the presence or absence of cyclin D1–CDK4 co-expression. (**B, C**) The indicated purified, recombinant proteins were incubated in the presence or absence of recombinant cyclin D1–CDK4 complex. ERK2 was used as a negative control. Post-incubation, differential phosphorylation was analyzed using phos-tag gels to detect mobility

*Figure 2 continued on next page*

*Figure 2 continued*

shifts. Immunoblot analysis was conducted to verify the presence of the recombinant proteins as indicated. (**D**) HEK293T cells transfected with either the indicated Flag-tagged proteins or an empty vector (EV). Cell lysates were subjected to immunoprecipitation followed by immunoblot analysis. Inputs (5%) represent whole-cell extracts before pull down. (**E**) Schematic representation of candidate CDK phosphorylation sites in GTSE1, as indicated by the PhosphoSitePlus database (*Hornbeck et al., 2015*). (**F**) HEK293T cells were transiently transfected with the indicated HA-GTSE1 mutants, in the presence or absence of cyclin D1–CDK4 co-expression. Changes in protein migration were analyzed by phos-tag gels. (**G**) Immunoblot and phos-tag gel analysis displaying cell cycle synchronization effects following 72 hr of serum starvation in parental T98G cells and AMBRA1 KO T98G cells. Cells were collected at various time points post-serum re-supplementation as indicated, followed by immunoblotting with the indicated antibodies. (**H**) HA-tagged wild-type GTSE1 (left) or GTSE1 mutated at positions 91, 261, 454, and 724 (Tetra SA) (right) was subjected to co-expression with various cyclins and CDKs, in the presence or absence of the indicated kinase inhibitors to observe the impact on phosphorylation. (**I**) Box plot showing GTSE1 phosphorylation levels normalized to total GTSE1 protein abundance across a pan-cancer cohort relative to adjacent normal tissue, utilizing data from the Clinical Proteomic Tumor Analysis Consortium (CPTAC) (*Varadarajan et al., 2020*). Statistical significance was assessed using the Wilcoxon rank-sum test, with a p-value threshold of <0.05. (**J**) Heatmap illustrating differential abundance in cancer of the three GTSE1 phosphorylation sites found in the current study using CPTAC data (*Varadarajan et al., 2020*). Color intensity reflects the log2-transformed Z-scores of identified GTSE1 phospho-peptides, indicating relative phosphorylation levels across various tumor cohorts compared to adjacent normal tissues. The complete analysis of all phosphorylation sites can be found in *Figure 3—figure supplement 1D*.

The online version of this article includes the following source data and figure supplement(s) for figure 2:

**Source data 1.** Original files for western blot analysis displayed in *Figure 2*.

**Source data 2.** PDF file containing original western blots for *Figure 2*, indicating the relevant bands.

**Source data 3.** The raw data for the box plot in *Figure 2I* depicting GTSE1 phosphorylation levels normalized to total GTSE1 protein abundance across a pan-cancer cohort is provided as log2(Intensities) across samples for global proteins and phospho-proteins of interest.

**Source data 4.** The pre-processed data for the box plot in *Figure 2I* and the heatmap illustrating differential abundance of GTSE1 phosphorylation sites in *Figure 2J* is provided as Row Z-score of GTSE1 log2(Intensities) across samples for global protein and phospho-protein.

**Figure supplement 1.** Evolutionary conservation analysis of GTSE1 phosphorylation sites across species.

mutants, only the S454A mutation demonstrated a complete abrogation of any shift in phos-tag gels (*Figure 2F*). In contrast, a single S454A mutation showed only a minor effect on GTSE1 shift (data not shown). These studies suggest that four major sites (S91, S262, S454, and S724) are phosphorylated (either directly and/or indirectly) in a cyclin D1–CDK4-dependent manner.

To gain an insight into the cell cycle pattern of GTSE1 abundance and phosphorylation status, we synchronized T98G cells (both parental and AMBRA1 KO pooled clones) by serum starvation followed by serum re-addition. Parental cells displayed a fluctuation in GTSE1 abundance, with low levels in G1 that subsequently increased in S and G2 phases (*Figure 3—figure supplement 1B*), in agreement with the literature (*Collavin et al., 2000*; *Monte et al., 2000*). In contrast, in AMBRA1 KO cells, GTSE1 levels were consistently elevated throughout the cell cycle, suggesting that elevated cyclin D1 levels are associated with an increased abundance of GTSE1 (*Figure 3—figure supplement 1B*). Additionally, while phosphorylation of GTSE1 in the parental cell line peaked during the G2/M transition (28–32 hr after serum re-addition), AMBRA1 KO cells exhibited sustained phosphorylation of GTSE1 across all cell cycle phases (*Figure 2G*, *Figure 3—figure supplement 1B*). We also generated a phospho-specific antibody that recognized GTSE1 only when phosphorylated on Ser262 (*Figure 3—figure supplement 1C*). Using this antibody, we confirmed the data obtained with phos-tag gel indicating that in cells overexpressing cyclin D1–CDK4 and in AMBRA1 KO cells, GTSE1 is hyper-phosphorylated on Ser262 (*Figure 2G*, *Figure 3—figure supplement 1C*).

Next, we leveraged data from the Clinical Proteomic Tumor Analysis Consortium (CPTAC) to examine the relevance of the identified phosphorylation events in a clinical context. We observed an enrichment of GTSE1 phospho-peptides within a pan-cancer cohort as opposed to adjacent, corresponding normal tissues (*Figure 2I*), underscoring the potential role of GTSE1 phosphorylation in tumorigenesis. Upon analyzing single phosphorylated sites, we found that nine were statistically enriched in cancers. Of these, three sites (S262, S454, and S724) were identified as dependent on cyclin D1–CDK4 in our study (*Figure 2J*, *Figure 3—figure supplement 1D*). (Data on Ser91 are not present in CPTAC.) These results underscore the potential pathophysiological significance of GTSE1 phosphorylation by cyclin D1–CDK4 in cancer.

Considering GTSE1 being an established target of cyclin B1–CDK1 during mitosis (*Singh et al., 2021*), we aimed to elucidate the phosphorylation patterns of GTSE1 by various cyclin–CDK complexes operating at distinct cell cycle phases. Wild-type GTSE1 was found to be phosphorylated in HEK293T

cells upon overexpression of all cyclin–CDK pairs, with its shifts in phos-tag gels being abolished upon treatment with the corresponding specific CDK inhibitor (*Figure 2H*, left panel). However, when the quadruple (S91A/S262A/S454A/S724A) GTSE1 mutant, referred to as 'Tetra SA', was expressed in HEK293T cells, cyclin D1–CDK4 was unable to induce any shift, whereas the other three cyclin–CDK pairs still sustained phosphorylation (*Figure 2H*, right panel). This suggests a unique phosphorylation profile conferred by cyclin D1–CDK4 in GTSE1, distinct from that induced by other cyclin–CDK complexes.

We noticed an elevation in endogenous GTSE1 levels in AMBRA1 KO cells compared to the parental line (*Figure 3A*, *Figure 3—figure supplement 1B*) in agreement with the findings of the proteomic screen performed in AMBRA1 KO cells (*Chaikovsky et al., 2021*; *Figure 1B*). A similar increase was observed when overexpressing wild-type cyclin D1 and, even more, an AMBRA1-insensitive, stable mutant of cyclin D1 (T286A) (*Simoneschi et al., 2021*; *Figure 3A, B*), suggesting that the elevated GTSE1 levels are due to high D-type cyclins present in AMBRA1 KO cells. We also used HCT-116 cells harboring an endogenous fusion of AMBRA1 to a minimally constructed Auxin Inducible Degron (mAID) at the N-terminus (*Yesbolatova et al., 2020*). This system allows for rapid and inducible degradation of AMBRA1 upon addition of auxin, thereby minimizing compensatory cellular rewiring. Again, we observed an increase in GTSE1 levels upon acute ablation of AMBRA1 (i.e., in 8 hr) (*Figure 3B*). In all cases, the upregulation in GTSE1 abundance was rescued upon Palbociclib treatment (*Figure 3A, B*), suggesting that this event was a consequence of increased levels of D-type cyclins. We also conducted cycloheximide (CHX) chase assays in HCT-116 mAID-AMBRA1 cells to assess GTSE1 protein stability and degradation kinetics. The assays revealed that in the context of AMBRA1 depletion, GTSE1 exhibited a prolonged half-life and reduced degradation rate when compared to control cells (*Figure 3C*). A parallel half-life assessment in parental U2OS cells and two AMBRA1 KO U2OS clones corroborated the finding that GTSE1 is stabilized in the absence of AMBRA1 (*Figure 3D*).

To further dissect the impact of cyclin D1–CDK4-mediated phosphorylation on GTSE1 stability, we engineered a phospho-mimicking mutant, referred to as 'Tetra SD' with the four serine residues replaced with an aspartate at positions 91, 261, 454, and 724. To circumvent the variability of transient transfection, U2OS cells were stably transduced with retroviruses encoding GFP-tagged wild-type GTSE1, Tetra SA (phospho-deficient), or Tetra SD (phospho-mimic). Subsequent CHX chase experiments showed a slower degradation kinetics of the Tetra SD mutant compared to the Tetra SA mutant and wild-type GTSE1 (*Figure 3E, F*). These stable cell lines, expressing fluorescent GTSE1 variants, were further analyzed via time-lapse microscopy during a CHX chase to quantify protein degradation through the diminishing fluorescence intensity at different times. The Tetra SD mutant exhibited a statistically significant slower fluorescence decrease rate than both Tetra SA and WT, indicating reduced degradation (*Figure 3G, H*).

Finally, we treated U2OS cells with CHX together with inhibitors targeting different degradation systems: MG132 (a proteasome inhibitor), MLN4924 (a CRL inhibitor), or Bafilomycin A (an autophagy inhibitor) (*Figure 3I*). The stabilization of GTSE1 in the presence of MG132 and MLN4924, but not Bafilomycin A, indicates that GTSE1 degradation mainly occurs via the ubiquitin–proteasome system, specifically implicating the involvement of a CRL. However, GTSE1 does not appear to be a substrate of CRL4[AMBRA1], as indicated by the lack of a physical interaction between AMBRA1 and GTSE1 (*Figure 3—figure supplement 1E*), as well as the fact that the stabilization of GTSE1 in AMBRA1 KO cells is secondary to the CDK4/6 activity (*Figure 3A, B*).

Next, we delved into the potential implications of GTSE1 phosphorylation by the cyclin D1–CDK4 complex. Previous research has connected GTSE1 with fundamental cellular processes, such as cell proliferation (*Lei et al., 2023*; *Lai et al., 2021*) and cell migration and invasion (*Scolz et al., 2012*; *Wu et al., 2017*; *Li et al., 2021*). Leveraging data from CPTAC, we examined possible correlations between both GTSE1 protein levels (*Figure 4—figure supplement 1A*) and levels of phosphorylated GTSE1 (*Figure 4A*) with markers of cell proliferation and cell migration across various cancer types. GTSE1 abundance and phosphorylation exhibited a statistically significant positive correlation with several proteins integral to cell proliferation, including PCNA, KI67, and MCM2, across multiple cancer cohorts. Conversely, in contrast to data suggesting a role for GTSE1 in promoting cell migration (*Scolz et al., 2012*; *Lin et al., 2019*; *Lei et al., 2023*; *Wu et al., 2017*) GTSE1 levels and phosphorylation demonstrated a negative trend with proteins associated with cell migration and

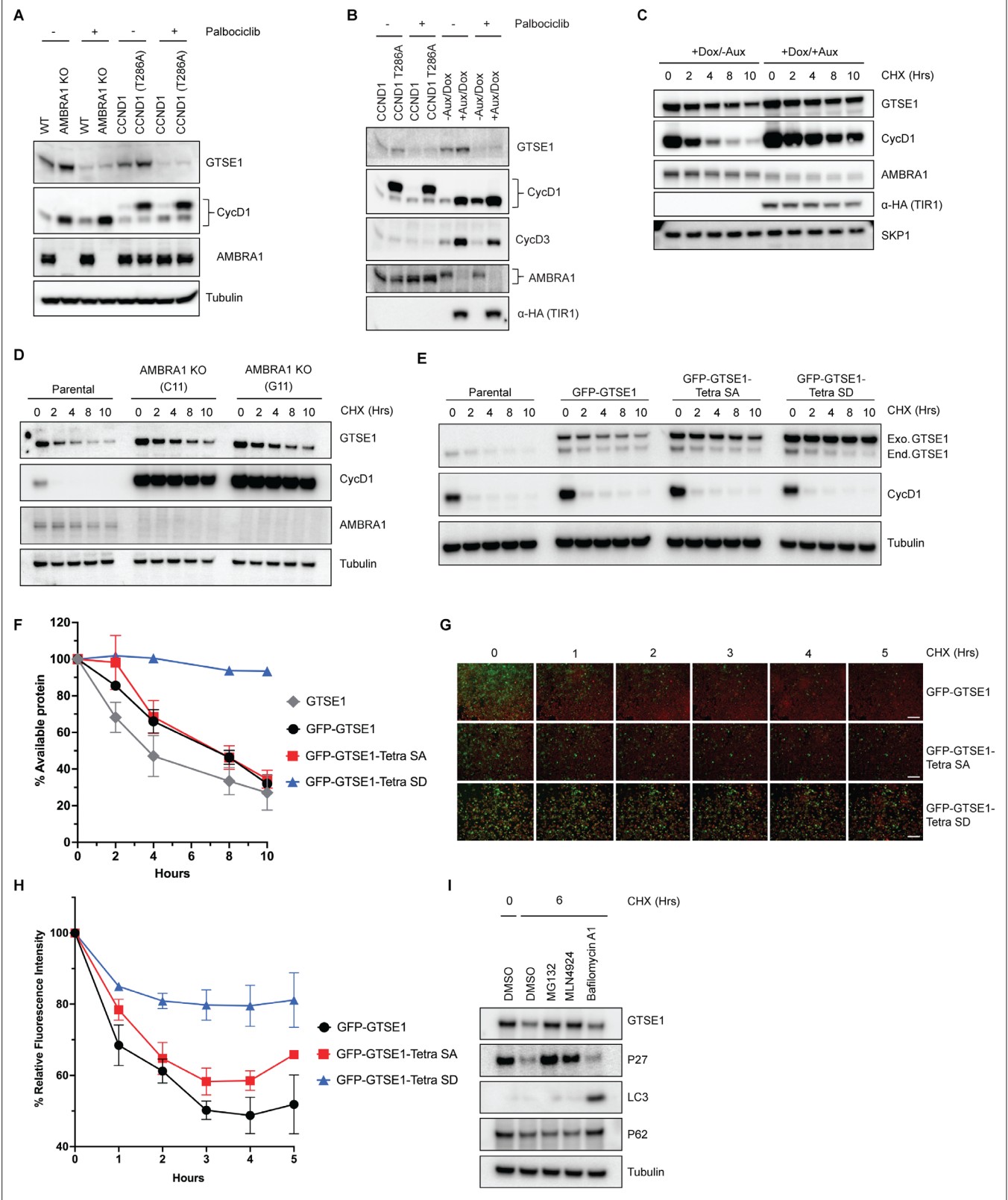

**Figure 3.** GTSE1 protein is stabilized upon its phosphorylation by cyclin D1–CDK4. (**A**) Immunoblot analysis of whole-cell extracts from parental HCT-116 cells, AMBRA1 knockout (KO) HCT-116 cells, and HCT-116 cells expressing either wild-type cyclin D1 or an AMBRA1-insensitive mutant of cyclin D1 (T286A), in the presence or absence of the CDK4/6 inhibitor Palbociclib. (**B**) Immunoblot analysis of whole-cell extracts from HCT-116 cells harboring an endogenous mini-AID domain fused to AMBRA1 N-terminus. Cells were subjected to either incubation with auxin and doxycycline to induce

*Figure 3 continued on next page*

*Figure 3 continued*

AMBRA1 degradation or transfection with cyclin D1(T286A) in the presence or absence of Palbociclib. (**C**) Cycloheximide (CHX) chase assay in HCT-116 cells with mAID-AMBRA1, treated with or without auxin and doxycycline to induce AMBRA1 degradation. Immunoblot analyses were conducted to assess the stability of GTSE1 and other indicated proteins, with the stable protein SKP1 used as a loading control. (**D**) Protein stability assessment via CHX chase in U2OS parental cells and AMBRA1 KO clones (C11 and G11). The protein levels of AMBRA1, GTSE1, and cyclin D1 were analyzed by immunoblotting, with tubulin serving as a loading control. (**E**) U2OS cells stably transduced with retroviruses expressing the indicated GTSE1 constructs were subjected to CHX chase assays. Subsequent immunoblotting was conducted for the indicated proteins, with tubulin utilized as a loading control. (**F**) Densitometric quantification of GTSE1 band intensity from (**E**) and two identical experiments normalized using tubulin. Initial band intensity at time 0 is set as the 100% reference point. Error bars represent SEM (*n* = 3 of biological replicates). (**G**) Time-lapse microscopy images of U2OS cells stably expressing EGFP-tagged the indicated GTSE1 constructs during a CHX chase. The EGFP signal intensity corresponds to GTSE1 levels, while cells are stained with a far-red cell tracker for cell masking. Scale bars indicate 120 µM. (**H**) Quantitative analysis of EGFP fluorescence intensity from time-lapse experiment shown in (**G**) plus two identical experiments. The initial fluorescence intensity was normalized to 100% at time zero. Data represent the mean fluorescence intensity from the three independent measurements, with error bars indicating SEM.(**I**) U2OS cells were treated with various inhibitors for 3 hr before harvest: the proteasome inhibitor MG132, the CRL inhibitor MLN4924, and the V-ATPase inhibitor Bafilomycin A1. Immunoblot analysis of the indicated proteins was performed, with tubulin as a loading control. p62 and LC3 serve as autophagy inhibition controls, while p27 is used as control for CRL- and ubiquitin–proteasome system (UPS)-dependent degradation.

The online version of this article includes the following source data and figure supplement(s) for figure 3:

**Source data 1.** Original files for western blot analysis displayed in *Figure 3*.

**Source data 2.** PDF file containing original western blots for *Figure 3*, indicating the relevant bands.

**Figure supplement 1.** Functional analysis of GTSE1 phosphorylation by cyclin D1-CDK4.

**Figure supplement 1—source data 1.** Original files for western blot analysis displayed in *Figure 3—figure supplement 1*.

**Figure supplement 1—source data 2.** PDF file containing original western blots for *Figure 3—figure supplement 1*, indicating the relevant bands.

**Figure supplement 1—source data 3.** The complete analysis for *Figure 3—figure supplement 1D* of all GTSE1 phosphorylation sites is provided as Row *Z*-score of GTSE1 log2(Intensities) across samples for global protein and phospho-protein.

epithelial–mesenchymal transition, such as Vimentin, MMP1, and ETS1, but these did not attain statistical significance.

To explore the influence of cyclin D1-mediated phosphorylation of GTSE1 on cellular phenotypes, we assessed the proliferation potential of AMBRA1 KO cells relative to their parental counterparts. AMBRA1 KO cells displayed an enhanced proliferation rate (*Figure 4—figure supplement 1B*), in agreement with the literature (*Simoneschi et al., 2021*). We then conducted comparable growth analyses using U2OS cells stably expressing various GTSE1 constructs. Cells stably expressing the Tetra SD mutant exhibited a higher proliferation rate compared to cells expressing either Tetra SA or WT GTSE1 (*Figure 4B*). Moreover, evaluations of cellular proliferation conducted using Cell Trace dye demonstrated that cells harboring the Tetra-SD mutant displayed an elevated proliferative index (*Figure 4C*, *Figure 4—figure supplement 1D, E*). This was evidenced by a greater number of cell generations and smaller fraction of non-divided cells relative to cells expressing WT GTSE1 and, even more so compared to cells expressing Tetra SA (*Figure 4C*, *Figure 4—figure supplement 1D, E*). Finally, cell cycle profiles showed that the expression of a phospho-mimicking GTSE1 mutant induces a decrease in the percentage of cells in G1 and an increase in the percentage of cells in S, similarly to what observed in AMBRA1 KO cells (*Figure 4—figure supplement 1F* and *Rona et al., 2024*).

## Summary

Cyclin D1 overexpression is a hallmark of many cancers, where it facilitates oncogenic transformation and progression. Amplifications and overexpression of the *CCND1* gene are seen in a diverse array of cancers, with amplifications in up to 15–20% of breast cancers and overexpression in over 50% of mantle cell lymphomas (*Musgrove et al., 2011*; *Cancer Genome Atlas Research, 2012*). The pervasiveness of cyclin D1 dysregulation in cancer highlights the critical need to understand its downstream effects. Despite this, the full spectrum of substrates and their impact on cellular function and oncogenesis has yet to be thoroughly investigated (*Kaulich et al., 2021*; *Sherr, 2000*; *Knudsen and Knudsen, 2008*; *Topacio et al., 2019*; *Tchakarska and Sola, 2020*). In the present study, we identified GTSE1, a pro-proliferative protein, as a novel substrate of the cyclin D1–CDK4 complex in tumor cells overexpressing cyclin D1 during G1 and subsequent phases. Our data indicate that cyclin D1–CDK4 is responsible for the phosphorylation of GTSE1 on four residues (S91, S262, S454, and S724). In contrast, cyclin A2–CDK2, cyclin E1–CDK2, and cyclin B1–CDK1 target additional and/

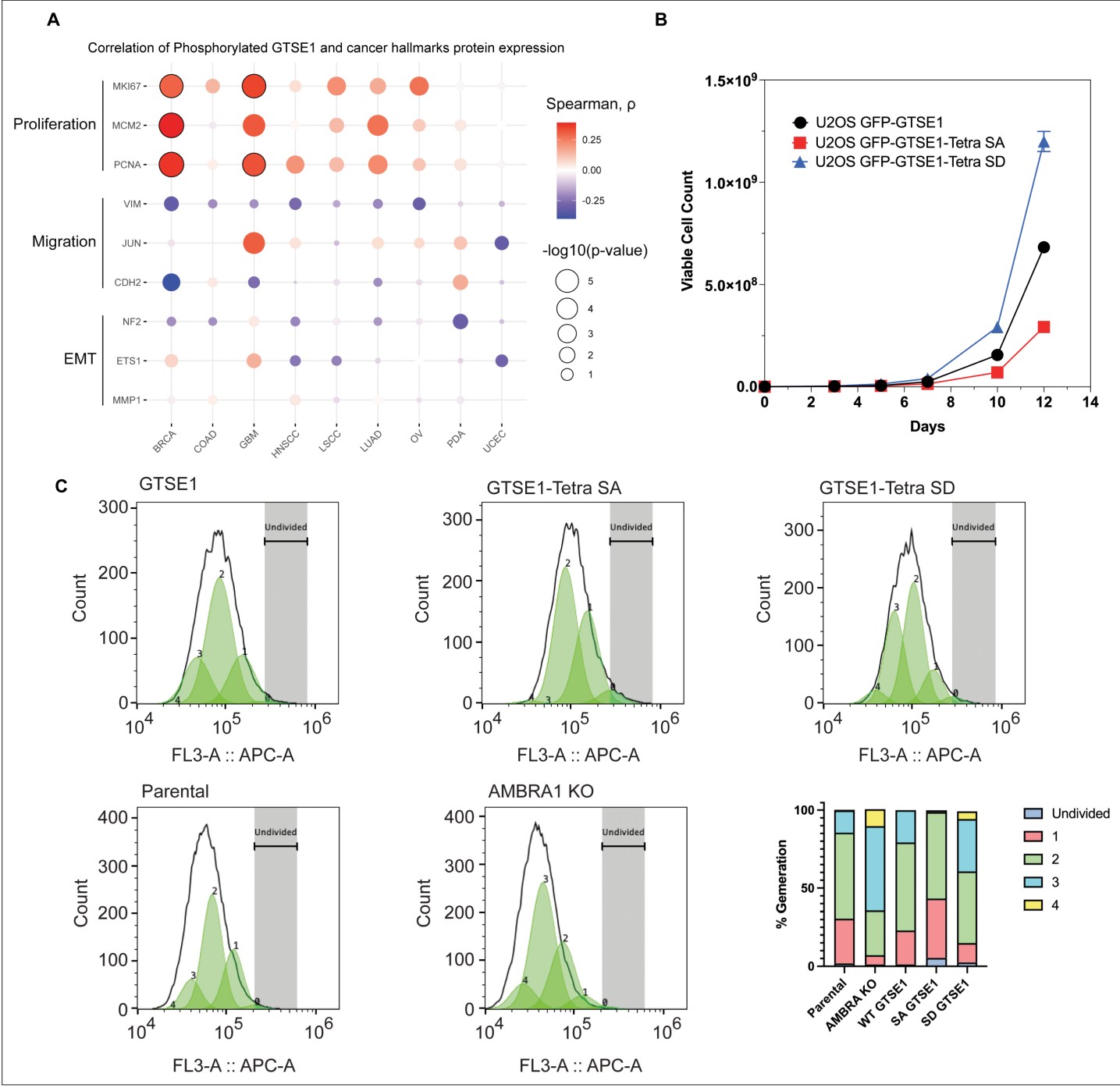

**Figure 4.** Increased cell proliferation upon cyclin D1–CDK4-mediated phosphorylation and stabilization of GTSE1. (**A**) Bubble plot derived from Clinical Proteomic Tumor Analysis Consortium (CPTAC) data analysis showing the correlation between the abundance of the bulk phospho-peptides of GTSE1 and the levels of proteins involved in cell proliferation or cell migration across various cancer types. The color intensity of each bubble represents the Spearman correlation coefficient, while the bubble size indicates the negative log10 transformation of the p-values. Bubbles outlined in black indicate statistically significant correlations with *q*-value <0.0001, adjusted by Benjamini–Hochberg (BH) method. (**B**) Growth curve analysis comparing the proliferation of U2OS cells stably expressing GFP-tagged wild-type GTSE1, GTSE1 Tetra SA (phospho-deficient), or GTSE1 Tetra SD (phospho-mimetic). GTSE1 Tetra SD cells proliferated significantly more than cells expressing either wild-type GTSE1 or GTSE1 Tetra SA. Statistical significance was determined using unpaired *T*-tests with p-values <0.05. Error bars represent the mean ± SEM (*n* = 3 per condition). (**C**) Proliferation analysis using CellTrace Far Red dye to assess the division of U2OS cells expressing wild-type GTSE1, Tetra SA, and Tetra SD, alongside a comparison between parental and AMBRA knockout (KO) U2OS cells. Following staining, the dye dilutes progressively with each cell division, allowing for the distinction of successive generations by the relative decrease in fluorescence intensity. Fluorescence-activated cell sorting was employed to accurately measure

*Figure 4 continued on next page*

*Figure 4 continued*

the dye dilution and thereby quantify the discrete populations representing each generation of the cell line. The gating strategy was implemented to identify live, single cells positive for GFP, as detailed in *Figure 4—figure supplement 1C*. The percentages of cells in each generation are as follows: Parental (Undivided: 1.9%, Gen 1: 28.6%, Gen 2: 55.0%, Gen 3: 14.3%, Gen 4: 0.46%), AMBRA KO (Undivided: 1.2%, Gen 1: 5.8%, Gen 2: 28.77%, Gen 3: 54.0%, Gen 4: 10.8%), WT GTSE1 (Undivided: 1.2%, Gen 1: 21.7%, Gen 2: 56.36%, Gen 3: 20.7%, Gen 4: 0%), Tetra SA GTSE1 (Undivided: 5.4%, Gen 1: 37.8%, Gen 2: 55.5%, Gen 3: 0%, Gen 4: 1.2%), and Tetra SD GTSE1 (Undivided: 2.52%, Gen 1: 12.3%, Gen 2: 45.9%, Gen 3: 33.52%, Gen 4: 4.96%). See additional biological replicates in *Figure 4—figure supplement 1D, E*.

The online version of this article includes the following source data and figure supplement(s) for figure 4:

**Source data 1.** The source data for *Figure 4A* is provided as Pairwise GTSE1 (phospho) ~ Hallmark (protein) Spearman correlations from proteomic input.

**Figure supplement 1.** GTSE1 phosphorylation status influences cell proliferation and cell cycle progression in cancer.

**Figure supplement 1—source data 1.** The source data for the bubble plot in *Figure 4—figure supplement 1A* is provided as Pairwise GTSE1 (protein) ~ Hallmark (protein) Spearman correlations from proteomic input.

or different sites as shown by the fact that they still induced mobility shift of the 'Tetra SA' mutant in cultured cells. GTSE1 has been established as a substrate of mitotic cyclins (*Liu et al., 2010*; *Singh et al., 2021*; *Monte et al., 2000*), but we observed that in human cells, when D-type cyclins are stabilized in the absence of AMBRA1, GTSE1 becomes phosphorylated across the various cell cycle phases. Accordingly, overexpression of cyclin D1–CDK4 induce GTSE1 phosphorylation. Thus, we propose that GTSE1 is phosphorylated by CDK4 and CDK6 particularly in pathological states, such as cancers displaying overexpression of D-type cyclins. In turn, GTSE1 phosphorylation induces its stabilization, leading to increased levels that contribute to enhanced cell proliferation. It is also possible that GTSE1 is also recognized in specific cell types and in particular stage/s of embryogenesis, when cyclin D1 levels are elevated, and high cell proliferation is crucial for proper tissue formation (*Ter Huurne and Stunnenberg, 2021*). The prolonged half-life of GTSE1 consequent to cyclin D1–CDK4-mediated phosphorylation suggests a mechanism through which cells could modulate the abundance of a key regulatory protein without necessitating new protein synthesis. This post-translational regulation could be a critical determinant in the rapid response to mitogenic signals or environmental cues.

Our study further established a correlation between phosphorylation of GTSE1 and markers of cell proliferation, as well as poor prognosis in human cancers, providing a partial explanatory basis for the proliferative phenotype associated with elevated D-type cyclins observed in human tumors. Future studies are warranted to further dissect the molecular mechanisms underpinning GTSE1's contribution to the cancerous phenotype, potentially paving the way for novel cancer therapeutics.

## Materials and methods
### Cell culture and transduction
HEK293T, HCT-116, T98G, and U2OS cell lines were cultured under specific conditions suited to each cell type. HEK293T and T98G cells were cultured in DMEM (Dulbecco's modified Eagle medium), while HCT-116 and U2OS cells were maintained in McCoy's 5A medium. All media were supplemented with 10% fetal bovine serum (FBS) and 1% penicillin–streptomycin and cells were incubated at 37°C in a humidified atmosphere containing 5% $CO_2$.

For retroviral production, HEK293T cells were co-transfected with GTSE1 expression constructs along with VSVG and PAX packaging plasmids using Lipofectamine 3000 (Invitrogen) according to the manufacturer's instructions. Virus-containing supernatants were harvested 48 hr post-transfection, filtered through a 0.45-µm pore size filter, and used for transducing target HCT-116 and U2OS cells in the presence of 8 µg/ml Polybrene. Stably transduced cells were selected by sorting for EGFP-positive cells by cell sorter.

For transient transfection studies, HEK293T cells were transfected with various plasmids using PEI (Polysciences); HCT-116 and U2OS were transiently transfected with Lipofectamine 3000 (Thermo Fisher Scientific). Where indicated, 24 hr after transfection, cells were treated with MG132 or MLN4924 for 4 hr before collection, following a protocol optimized for maximal transfection efficiency and minimal cellular toxicity. Post-transfection, cells were maintained under standard culture conditions before being harvested for subsequent experiments.

## Plasmids

*Homo sapiens* cDNAs were amplified by PCR using KAPA HiFi DNA Polymerase (Kapa Biosystems) and sub-cloned into a variety of vector backbones, including modified pCDNA3.1 vectors containing N-terminal Flag or HA and pBABE-PURO retroviral vectors containing N-terminal eGFP. FFSS indicates a tandem 2×Flag–2×Strep tag. Site-directed mutagenesis was performed using KAPA HiFi DNA Polymerase (Kapa Biosystems).

## Western blotting and antibodies

Protein extracts were prepared using RIPA buffer supplemented with protease inhibitors (Complete ULTRA, Roche) and phosphatase inhibitors (PhosSTOP, Roche). The insoluble fraction was removed by centrifugation ($20,000 \times g$) for 15 min at 4°C. Protein concentrations in cell lysates were normalized using the Pierce BCA Protein Assay Kit (Thermo Fisher Scientific), according to the manufacturer's instructions. Proteins were separated by SDS–PAGE and transferred to PVDF membranes. Membranes were blocked with 5% non-fat dry milk and incubated with primary antibodies overnight at 4°C, followed by HRP-conjugated secondary antibodies. Enhanced chemiluminescence (Thermo Fisher Scientific) was used for detection.

For high-resolution separation of phosphorylated proteins, we used phos-tag gels with 7.5% acrylamide and 40 µM $Zn^{2+}$ (Fujifilm Wako Chemicals U.S.A.), except for the experiment in *Figure 2G* in which we used 6% acrylamide and 100 µM $Zn^{2+}$. Electrophoresis was performed at 50 V, according to the manufacturer's instructions. The following antibodies were used: GTSE1 (1:1000, Bethyl Laboratories #A302-274A), β-actin (1:5000, Sigma-Aldrich A5441), AMBRA1 (1:1000, Proteintech Group 13762-1-AP), cyclin A (1:5000, M.P. laboratory), cyclin B1 (1:5000, M.P. laboratory), cyclin D1 (1:1000, Abcam ab16663), p-cyclin D1 (T286) (1:1000, Cell Signaling Technology 3300S), cyclin E (1:1000, Santa Cruz Biotechnology sc-247), Flag (1:2000, Sigma-Aldrich F1804), Flag (1:2000, Sigma-Aldrich F7425), GST (1:5000, GE Healthcare 27457701), HA (1:2000, Bethyl Laboratories A190-108A), LC-3 (1:5000, Novus Biological NB100-2220), p21 (1:1,000, Cell Signaling Technology 2947S), p27 (1:1000, BD Biosciences 610241), p62 (1:5000, MBL International PM045), RB (1:1000, Cell Signaling Technology 9313S), RB (1:1000, Cell Signaling Technology 9309S), p-RB (S807/811) (1:1000, Cell Signaling Technology 9308S), SKP1 (1:5,000, M.P. laboratory), and α-tubulin (1:5000, Sigma-Aldrich T6074). A phospho-specific antibody against GTSE1 phosphorylated on Ser262 was generated by YenZym. Briefly, a peptide containing the phospho-epitope, which includes amino acids 254–269 (KPKKEI-PApSPSRTKIP) was synthesized. This peptide was then used to immunize rabbits, using Cys-KLH as immunogen, prompting the production of antibodies against the phospho-epitope. Following immunization, phospho-specific antibodies were purified by utilizing a phosphorylated peptide-conjugated affinity matrix. To ensure the specificity and efficacy of the antibody, an ELISA was performed on both pre- and post-purified serum.

## In vitro phosphorylation

Assays were carried out using 1 µg of recombinant substrate proteins in the presence or absence of 0.1 µg of kinase enzymes. Where indicated, 1 µM Palbociclib was added to reaction mixtures containing CDK4 and cyclin D1 and subjected to pre-incubation on ice for 10 min before the addition of the recombinant substrates. The phosphorylation reaction was conducted in kinase reaction buffer, containing 25 mM Tris-HCl (pH 7.5), 5 mM beta-glycerophosphate, 2 mM dithiothreitol, 0.1 mM $Na_3VO_4$, 10 mM $MgCl_2$, and 2 mM ATP. The reaction mixture was maintained at 30°C for a duration of 1 hr. To terminate the reaction, Laemmli buffer containing SDS and beta-mercaptoethanol was added to the samples, followed by heating at 95°C for 10 min.

## Chemicals and reagents

CHX, Bafilomycin A1, MG132, and MLN4924 were obtained from Sigma-Aldrich (United States). PF-06873600 and Palbociclib were purchased from Selleckchem (United States).

RO-3306 was acquired from Roche (United States). Auxin (Indole-3-acetic acid) and doxycycline were sourced from Thermo Fisher Scientific (United States). Rb (Retinoblastoma Human Recombinant) fused with a 6X His tag, containing the C-terminal 792–928 aa, was obtained from Raybiotech (United States). GTSE1, full-length, with N-terminal GST and C-terminal HIS tag, was sourced from

Origene (United States). GST-tagged ERK2 was acquired from Abcam (United States). His-tagged Human Cyclin D1 and CDK4 were obtained from LSBio (United States).

## Growth curve analysis

Cell proliferation was assessed by plating cells at a fixed density and counting at specific intervals using Vi-CELL BLU Cell Viability Analyzer (Beckman Coulter). For cell count normalization, Trypan Blue exclusion was used to assess cell viability.

## Cell proliferation analysis

Cells were labeled with the CellTrace Far Red dye (Invitrogen) according to the manufacturer's instructions, ensuring that the initial fluorescence intensity was homogenous across the cell population. This method relies on the principle of dye dilution to trace multiple generations of cells through flow cytometry. Following labeling, the cells were cultured under appropriate conditions to allow for cell division (confluency <70%). Following five generation time, samples were collected and subjected to flow cytometric analysis. The progressive halving of the fluorescent dye intensity, as a result of cell division, was monitored, enabling the quantification of cell generations.

## Cell cycle studies

T98G cells were synchronized by serum starvation. Subconfluent plates of cells were trypsinized washed with PBS, and replated in DMEM containing 0.01% FBS. Cells were kept in this medium for 72 hr before they were trypsinized and replated at 70% confluency in DMEM containing 10% FBS to allow for cell cycle re-entry. Cells were collected at various time points following serum re-addition by scraping. Cell cycle was also monitored through EdU and propidium iodide staining and FACS analysis following the instructions of the manufacture.

## Time lapse fluorescence monitoring

For the monitoring of GFP-tagged GTSE1 degradation, U2OS cells were plated at a density of 50,000 cells per well within 96-well plates. Following a period of 24 hr from plating, the cells were subjected to live cell imaging using the Cytation 5 Cell Imaging Reader (BioTek, Winooski, VT), which is equipped with a dedicated module for real-time cell analysis. The imaging apparatus maintained the cells in an optimal environment, regulated at 37°C with 5% $CO_2$ levels. A Far Red cell tracer dye was applied to the cells prior to imaging, providing a distinct cellular outline for accurate identification during subsequent analysis. Images were captured at predetermined intervals following treatment with CHX. Fluorescent emissions from both EGFP and Far Red spectra were monitored. Post-acquisition, images were analyzed to measure the cumulative EGFP fluorescent intensity within the confines delineated by the Far Red cellular demarcation. All procedures were conducted in triplicates.

## Bioinformatic analysis

Bioinformatic analysis was performed using R statistical packages. For the visualization of tables, the 'formattable' package in R was employed. Differential proteomic profiling comparing parental U2OS cells to AMBRA1 KO U2OS cells, utilized raw data from a shotgun proteomic screen previously published (*Chaikovsky et al., 2021*). Proteins differentially expressed with an adjusted p-value <0.05 were ordered according to the log2 fold change (Log2FC) comparing AMBRA1 KO to parental cells. The significance of the differentially expressed proteins was determined by the false discovery rate method. An annotated list of phosphorylated proteins that contain the canonical CDK phosphorylation site [S/T*]PX[K/R] utilized the PhosphoSitePlus database (*Hornbeck et al., 2015*). This list was integrated with the proteomic data from the prior screen by matching gene names, to identify potential CDK substrates altered in the AMBRA1 KO context. The bioinformatic analysis for assessing the differential abundance and phosphorylation levels of GTSE1, as well as the correlation of GTSE1 abundance with different protein signatures in various cancer cohorts was conducted utilizing data from the CPTAC database (*Ellis et al., 2013*).

## Acknowledgements

We thank Dr. Gregory David for critically reading the manuscript. MP is thankful to TM Thor and TB Balduur for their continuous support. This work was supported by NIH GM136250 to MP. MP is an

investigator with the Howard Hughes Medical Institute. SK is a recipient of the Life Sciences Research Foundation (LSRF) Postdoctoral Fellowship and has been an EMBO Long Term Postdoctoral Fellow. NGV and TJGR are thankful for NIH Institutional Training Grant (T32GM136542). TJGR thank HHMI for Gilliam Fellowship (GT15758) support.

## Additional information

### Competing interests

YeonTae Jeong: is currently an employee of Arvinas, Inc, contributions to this paper were completed before joining Arvinas. Michele Pagano: is or has been an advisor for and has financial interests in SEED Therapeutics, Triana Biomedicines, CullGen, Kymera Therapeutics, Serinus Biosciences, and Umbra Therapeutics. The other authors declare that no competing interests exist.

### Funding

| Funder | Grant reference number | Author |
|---|---|---|
| National Institutes of Health | GM136250 | Michele Pagano |
| Howard Hughes Medical Institute | | Michele Pagano |
| Life Sciences Research Foundation | | Sharon Kaisari |
| Howard Hughes Medical Institute | GT15758 | Tania J González-Robles |
| National Institutes of Health | T32GM136542 | Nelson García-Vázquez Tania J González-Robles |
| European Molecular Biology Organization | | Sharon Kaisari |

The funders had no role in study design, data collection, and interpretation, or the decision to submit the work for publication.

### Author contributions

Nelson García-Vázquez, Formal analysis, Investigation; Tania J González-Robles, Data curation, Visualization; Ethan Lane, Daria Spasskaya, Qingyue Zhang, Marc A Kerzhnerman, YeonTae Jeong, Marta Collu, Daniele Simoneschi, Gergely Róna, Investigation; Kelly V Ruggles, Resources, Visualization; Sharon Kaisari, Conceptualization, Supervision, Investigation, Writing – original draft, Writing – review and editing; Michele Pagano, Conceptualization, Funding acquisition, Writing – original draft, Writing – review and editing

### Author ORCIDs

Tania J González-Robles  https://orcid.org/0000-0001-9292-382X
Daniele Simoneschi  https://orcid.org/0000-0003-0448-4295
Kelly V Ruggles  https://orcid.org/0000-0002-0152-0863
Sharon Kaisari  https://orcid.org/0000-0002-6884-3886
Michele Pagano  https://orcid.org/0000-0003-3210-2442

Reviewer #1 (Public review): https://doi.org/10.7554/eLife.101075.3.sa1
Reviewer #2 (Public review): https://doi.org/10.7554/eLife.101075.3.sa2
Reviewer #3 (Public review): https://doi.org/10.7554/eLife.101075.3.sa3
Author response https://doi.org/10.7554/eLife.101075.3.sa4

# Additional files

## Supplementary files
MDAR checklist

## Data availability
This study primarily analyzed publicly available datasets from the Clinical Proteomic Tumor Analysis Consortium (CPTAC). The CPTAC datasets used in this study include pan-cancer proteomic data referenced for the analysis of GTSE1 phosphorylation levels across multiple cancer types (*Figure 2I and J*, *Figure 4A*, *Figure 3—figure supplement 1D*, and *Figure 4—figure supplement 1A*). All analytical results generated during this study are included in the manuscript and supporting files. Source data files have been provided for the relevant figures. Uncropped gel scans used for figure generation have been deposited in Mendeley Data Repository (https://doi.org/10.17632/xzkw7hrwjr.1). Plasmids generated in this study have been deposited in Addgene. Cell lines and the phospho-specific antibody against GTSE1 phosphorylated on Ser262 generated in this study are available from the corresponding authors upon reasonable request. All other materials, reagents, and computational tools described in this paper are available from the authors or from standard commercial sources as specified in the Materials and methods section. **CPTAC data access**: The CPTAC datasets used in this study are available through the CPTAC Data Portal (https://pdc.cancer.gov/pdc/cptac-pan-cancer) and the Genomic Data Commons (https://portal.gdc.cancer.gov/). Specifically, we used the pre-processed CPTAC datasets Proteome_UMich_SinaiPreprocessed_GENECODE34_v1.zip and Phosphoproteome_UMich_SinaiPreprocessed_GENECODE34_v1.zip. Access to the raw datasets requires approval through the dbGaP authorization process (https://dbgap.ncbi.nlm.nih.gov/), in accordance with the data access policies established by the National Cancer Institute. **CPTAC data citation**: In accordance with CPTAC data usage policies, we have cited the appropriate CPTAC publications and acknowledged the CPTAC consortium as the source of primary data. Specifically, we used the following CPTAC datasets: Pan-cancer proteomic and phosphoproteomic data as referenced in *Ellis et al., 2013*, and The Cancer Genome Atlas (TCGA) data for transcriptomic analysis as referenced in *The Cancer Genome Atlas Research Network et al., 2013*. The proteomic data from AMBRA1 knockout U2OS cells compared to parental U2OS cells referenced in *Figure 1A–C* was obtained from previously published work by *Chaikovsky et al., 2021*. This dataset, titled "Shotgun mass spectrometry analysis comparing AMBRA1 KO U2OS cell populations and wild-type controls," is available in the ProteomeXchange Consortium via the PRIDE partner repository with the dataset identifier PXD021789.

The following dataset was generated:

| Author(s) | Year | Dataset title | Dataset URL | Database and Identifier |
|---|---|---|---|---|
| Pagano M | 2025 | Stabilization of GTSE1 by cyclin D1-CDK4/6-mediated phosphorylation promotes cell proliferation: relevance in cancer prognosis | https://doi.org/10.17632/xzkw7hrwjr.1 | Mendeley Data, 10.17632/xzkw7hrwjr.1 |

The following previously published dataset was used:

| Author(s) | Year | Dataset title | Dataset URL | Database and Identifier |
|---|---|---|---|---|
| Chaikovsky AC | 2021 | Shotgun mass spectrometry analysis comparing AMBRA1 KO U2OS cell populations and wild-type controls | https://dx.doi.org/10.6019/PXD021789 | ProteomeXchange, 10.6019/PXD021789 |

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
