## [Editor Report · eLife Assessment]

In this article, García-Vázquez et al. report **valuable** findings demonstrating that G2 and S phases expressed protein 1 (GTSE1), is a previously unappreciated non-pocket substrate of the cyclin D/cyclin-dependent kinase (CDK) 4/6 axis. The authors provide **convincing** evidence showing that this mechanism is triggered in pathological states in which cyclin D levels are elevated (e.g., cancer). Overall, this study holds a promise to improve understanding of the mechanisms underpinning cell cycle progression including its dysregulation in neoplasia and may thus be of broad interest to researchers belonging to diverse biomedical disciplines ranging from cancer research to cell biology.

---

## [Referee Report · Reviewer #1 (Public review)]

Summary:

García-Vázquez et al. identify GTSE1 as a novel target of the cyclin D1-CDK4/6 kinases. The authors show that GTSE1 is phosphorylated at four distinct serine residues and that this phosphorylation stabilizes GTSE1 protein levels to promote proliferation. This regulatory link appears to be particularly important in pathological conditions such as cancer, where cyclin D levels are elevated.

Strengths:

The authors support their findings with several previously published results, including databases. In addition, the authors perform a wide range of experiments to support their findings.

Impact:

The authors reveal a mechanism by which elevated levels of cyclin D1-CDK4 can stabilize GTSE1 throughout the cell cycle via phosphorylation. This provides insight into the role of cyclin D1-CDK4 in regulating the cell cycle and promoting cancer growth.

Comments on revisions:

The authors have addressed all my concerns, and I would like to thank them for their efforts on this great study.

---

## [Referee Report · Reviewer #2 (Public review)]

Summary:

The manuscript by García-Vázquez et al identifies the G2 and S phases expressed protein 1(GTSE1) as a substrate of the CycD-CDK4/6 complex. CycD-CDK4/6 is a key regulator of the G1/S cell cycle restriction point, which commits cells to enter a new cell cycle. This kinase is also an important therapeutic cancer target by approved drugs including Palbocyclib. Identification of substrates of CycD-CDK4/6 can therefore provide insights into cell cycle regulation and the mechanism of action of cancer therapeutics. A previous study identified GTSE1 as a target of CycB-Cdk1 but this appears to be the first study to address the phosphorylation of the protein by Cdk4/6.

The authors identified GTSE1 by mining an existing proteomic dataset that are elevated in AMBRA1 knockout cells. The AMBRA1 complex normally targets D cyclins for degradation. From this list they then identified proteins that contain a CDK4/6 consensus phosphorylation site and were responsive to treatment with Palbocyclib.

The authors show CycD-CDK4/6 overexpression induces a shift in GTSE1 on phostag gels that can be reversed by Palbocyclib. In vitro kinase assays also showed phosphorylation by CDK4. The phosphorylation sites were then identified by mutagenizing the predicted sites and phostag gets to see which eliminated the shift.

The authors go on to show that phosphorylation of GTSE1 affects the steady state level of the protein. Moreover, they show that expression and phosphorylation of GTSE1 confer growth advantage on tumor cells and correlate with poor prognosis in patients.

Strengths:

The biochemical and mutagenesis evidence presented convincingly show that the GTSE1 protein is indeed a target of the CycD-CDK4 kinase. The follow-up experiments begin to show that the phosphorylation state of the protein affect function and have an impact on patient outcome.

Weaknesses:

It is not clear at which stage in the cell cycle GTSE1 is being phosphorylated and how this is affecting the cell cycle. Considering that the protein is also phosphorylated during mitosis by CycB-Cdk1, it is unclear which phosphorylation events may be regulating the protein.

Additional comments for the revised manuscript

The authors have made many modifications to the manuscript in response to the reviewer comments, including the addition of new data that have clarified some of the conclusions. Some of the questions regarding the phase of the cell cycle affected have been addressed with flow cytometry.

There is one issue raised in the first review that can be better addressed. As the authors mentioned in their rebuttal letter, all the reviewers and editor concluded from the original manuscript that GTSE1 was being proposed as a physiological target of CycD-Cdk4 even in non-transformed cells. The authors believe that GTSE1 is likely only a target in cancerous cells that overexpress CycD and have made alterations in the abstract and main text making this point more clear.

Some additional evidence that GTSE1 phosphorylation is occurring in CycD overexpressing tumor cells would strengthen this argument beyond the overexpression experiments presented in the manuscript. For example, in Supplemental Fig 4A of the revised manuscript, bubble plots from CPTAC data is used to show that total protein levels of GTSE1 correlate with proteins associated with proliferation and metastasis. Do levels of GTSE1 correlate with CycD in this data set?

---

## [Referee Report · Reviewer #3 (Public review)]

Summary:

This paper identifies GTSE1 as a substrate of cyclin D1-CDK4/6 complexes when cyclin D1 is significantly over-expressed (as is common in cancers) rather than its endogenous level. GTSE is stabilized by phosphorylation and GTSE1 correlates with cancer prognosis, probably through an effect on cell proliferation.

Strengths:

There are few bonafide cyclin D1-Cdk4/6 substrates identified to be important in vivo so GTSE1 represents a potentially important finding for the field. Currently, the only cyclin D1 substrates involved in proliferation are the Rb family proteins.

Weaknesses:

GTSE1 is not a 'normal' target of cyclin D1-Cdk4/6, but rather only a target in a pathological situation.

---

## [Author Response]

The following is the authors’ response to the original reviews

**eLife Assessment**
In this valuable study, García-Vázquez et al. provide solid evidence suggesting that G2 and S phases expressed protein 1 (GTSE1), is a previously unappreciated non-pocket substrate of cyclin D1-CDK4/6 kinases. To this end, this study holds a promise to significantly contribute to an improved understanding of the mechanisms underpinning cell cycle progression. Notwithstanding these clear strengths of the article, it was thought that the study may benefit from establishing the precise role of cyclin D1-CDK4/6 kinase-dependent GTSE1 phosphorylation in the context of cell cycle progression, …

We do not claim, as editors and reviewers appear to have interpreted, that GTSE1 is phosphorylated by cyclin D1-CDK4 in the G1 phase of the cell cycle under normal physiologic conditions. Indeed, we agree with the existing literature indicating that in cells that do not express high levels of cyclin D1, GTSE1 is expressed predominantly during S and G2 phase (hence the name GTSE1, which stands for G-Two and S phases expressed protein 1) and is phosphorylated by mitotic cyclins in early mitosis. Even during G1, when the levels of cyclin D1 peak, GTSE1 is not phosphorylated in normal cells. This could be due to either a higher affinity between GTSE1 and mitotic cyclins as compared to D-type cyclins or to a higher concentration of mitotic cyclins compared to D-type cyclins. In the current manuscript, we show that higher levels of cyclin D1 can drive the sustained phosphorylation of GTSE1 across all cell cycle points. To reach this conclusion, we do not rely only on the overexpression of exogenous cyclin D1. In fact, we observe similar effect when we deplete endogenous AMBRA1, resulting in the stabilization of endogenous cyclin D1 in all cell cycle phases (see Figure 2G and Figure supplement 3B). As we had already mentioned in the Discussion section, we propose that GTSE1 is phosphorylated by CDK4 and CDK6 particularly in pathological states, such as cancers displaying overexpression of D-type cyclins (i.e., it is possible that the overexpression overcomes the lower affinity of the cyclin D-GTSE1 complex). In turn, phosphorylation of GTSE1 induces its stabilization, leading to increased levels that, as expected based on the existing literature, contribute to enhanced cell proliferation. So, the role of the cyclin D1-CDK4/6 kinase-dependent GTSE1 phosphorylation is to stabilize GTSE1 independently of the cell cycle. In sum, our study suggests that overexpression of cyclin D1, which is often observed in cancers cells beyond the G1 phase, induces phosphorylation of GTSE1 at all points in the cell cycle.

… obtaining more direct evidence that cyclin D1-CDK4/6 kinase phosphorylate indicated sites on GTSE1 (e.g., S454) …

We show that treatment of cells with palbociclib completely abolished the effect of cyclin D1-CDK4 on the GTSE1 shift observed using Phos-tag gels (Figure 2H). Moreover, mutagenesis analysis shows that S91, S262, and S724 are phosphorylated in a cyclin D1-CDK4-dependent manner (Figure 2F and Figure supplement 3A). Compared to wild-type GTSE1, a triple mutant (S91A/S262A/S724A) displayed loss of slower-migrating bands upon co-expression of cyclin D1-CDK4, suggesting diminished phosphorylation. Nevertheless, a residual slow-migrating band persisted, prompting further mutations of the triple GTSE1 mutant in S331 and S454 (individually), which do not have a CDK-phosphorylation consensus, but were identified in several published phospho-proteomics studies. From these two quadruple mutants, only the that containing the S454A mutation demonstrated a complete abrogation of any shift in phos-tagTM gels (Figure 2F). These studies suggest that four major sites (S91, S262, S454, and S724) are phosphorylated (either directly and/or indirectly) in a cyclin D1-CDK4-dependent manner.

… and mapping a degron in GTSE1 whose function may be blocked by cyclin D1-CDK4/6 kinase-dependent phosphorylation.

We show that stabilization or overexpression of cyclin D1, which is often observed in human cancers, promotes GTSE1 phosphorylation on S91, S262, S454, and S724, resulting in GTSE1 stabilization. Similarly, a phospho-mimicking mutant with the 4 serine residues replaced with an aspartate at positions 91, 261, 454, and 724 display increased half-life. While we appreciate the editor’s suggestion and agree on these being interesting questions, we would like to respectfully point out that mapping the GTSE1 degron and understanding how it is affected by cyclin D1-CDK4/6-dependent phosphorylation is outside the scope of the current project and will require an extensive set of experiments and tools. Accordingly, the three reviewers did not ask to map the GTSE1 degron. We plan on addressing these interesting questions as part of a follow-up study.

**Reviewer #1 (public review):**
Summary:García-Vázquez et al. identify GTSE1 as a novel target of the cyclin D1-CDK4/6 kinases. The authors show that GTSE1 is phosphorylated at four distinct serine residues and that this phosphorylation stabilizes GTSE1 protein levels to promote proliferation.Strengths:The authors support their findings with several previously published results, including databases. In addition, the authors perform a wide range of experiments to support their findings.Weaknesses:I feel that important controls and considerations in the context of the cell cycle are missing. Cyclin D1 overexpression, Palbociclib treatment and apparently also AMBRA1 depletion can lead to major changes in cell cycle distribution, which could strongly influence many of the observed effects on the cell cycle protein GTSE1. It is therefore important that the authors assess such changes and normalize their results accordingly.

We have approached the question of GTSE1 phosphorylation to account for potential cell cycle effects from multiple angles:

(i) We conducted in vitro experiments with purified, recombinant proteins and shown that GTSE1 is phosphorylated by cyclin D1-CDK4 in a cell-free system (Figure 2A-C). These experiments provide direct evidence of GTSE1 phosphorylation by cyclin D1-CDK4 without the influence of any other cell cycle effectors.

(ii) We present data using synchronized AMBRA1 KO cells (new Figure 2G and Figure supplement 3B). In agreement with what we had shown previously (Simoneschi et al., Nature 2021, PMC8875297), AMBRA1 KO cells progress faster in the cell cycle but they are still synchronized as shown, for example, by the mitotic phosphorylation of Histone H3, peaking at 32 hours after serum readdition like in parental cells. Under these conditions we observed that while phosphorylation of GTSE1 in parental cells is evident in the last two time points, AMBRA1 KO cells exhibited sustained phosphorylation of GTSE1 across all cell cycle phases. This was evident enough when using Phos-tag gels as in the top panel of the old Figure 2G. We now re-run one the biological triplicates of the synchronized cells using higher concentration of Zn^+2^-Phos-tag reagent and lower voltage to allow better separation of the phosphorylated bands. Under these conditions, GTSE1 phosphorylation is better appreciable (top panel of the new Figure 2G). This experiment provides evidence that high levels of cyclin D1 in AMBRA1 KO cells affect GTSE1 phosphorylation independently of the specific points in the cell cycle.

(iii) The relative short half-life of GTSE1 (<4 hours) makes its levels sensitive to acute treatments such as Palbociclib or acute AMBRA1 depletion. The effects of these treatments on GTSE1 levels are measurable within a time frame too short to significantly affect cell cycle progression. For example, we used cells with fusion of endogenous AMBRA1 to a mini-Auxin Inducible Degron (mAID) at the N-terminus. This system allows for rapid and inducible degradation of AMBRA1 upon addition of auxin, thereby minimizing compensatory cellular rewiring. Again, we observed an increase in GTSE1 levels upon acute ablation of AMBRA1 (i.e., in 8 hours) (Figure 3B), when no significant effects on cell cycle distribution are observed (please see Simoneschi et al., Nature 2021, PMC8875297 and Rona et al., Mol. Cell 2024, PMC10997477).

Altogether, the above lines of evidence support our conclusion that GTSE1 is a target of cyclin D1-CDK4, independent of cell cycle effects.

In conclusion, we do not claim that GTSE1 is phosphorylated by cyclin D1-CDK4 in the G1 phase of the cell cycle under normal physiologic conditions. Indeed, we agree with the existing literature indicating that in cells that do not express high levels of cyclin D1, GTSE1 is expressed predominantly during S and G2 phase (hence the name GTSE1, which stands for G-Two and S phases expressed protein 1) and is phosphorylated by mitotic cyclins in early mitosis. Even during G1, when the levels of cyclin D1 peak, GTSE1 is not phosphorylated in normal cells. This could be due to either a higher affinity between GTSE1 and mitotic cyclins as compared to D-type cyclins or to a higher concentration of mitotic cyclins compared to D-type cyclins. In the current manuscript, we show that higher levels of cyclin D1 can drive the sustained phosphorylation of GTSE1 across all cell cycle points. To reach this conclusion, we do not rely only on the overexpression of exogenous cyclin D1. In fact, we observe similar effect when we deplete endogenous AMBRA1, resulting in the stabilization of endogenous cyclin D1 in all cell cycle phases (see Figure 2G and Figure supplement 3B). As we had already mentioned in the Discussion section of the original submission, we propose that GTSE1 is phosphorylated by CDK4 and CDK6 particularly in pathological states, such as cancers displaying overexpression of D-type cyclins (i.e., it is possible that the overexpression overcomes the lower affinity of the cyclin D1-GTSE1 complex). In turn, phosphorylation of GTSE1 induces its stabilization, leading to increased levels that, as expected based on the existing literature, contribute to enhanced cell proliferation. In sum, our study suggests that overexpression of cyclin D1, which is often observed in cancers cells beyond the G1 phase, induces phosphorylation of GTSE1 at all points in the cell cycle.

**Reviewer #2 (public review):**
Summary:The manuscript by García-Vázquez et al identifies the G2 and S phases expressed protein 1(GTSE1) as a substrate of the CycD-CDK4/6 complex. CycD-CDK4/6 is a key regulator of the G1/S cell cycle restriction point, which commits cells to enter a new cell cycle. This kinase is also an important therapeutic cancer target by approved drugs including Palbocyclib. Identification of substrates of CycD-CDK4/6 can therefore provide insights into cell cycle regulation and the mechanism of action of cancer therapeutics. A previous study identified GTSE1 as a target of CycB-Cdk1 but this appears to be the first study to address the phosphorylation of the protein by Cdk4/6.The authors identified GTSE1 by mining an existing proteomic dataset that is elevated in AMBRA1 knockout cells. The AMBRA1 complex normally targets D cyclins for degradation. From this list, they then identified proteins that contain a CDK4/6 consensus phosphorylation site and were responsive to treatment with Palbocyclib.The authors show CycD-CDK4/6 overexpression induces a shift in GTSE1 on phostag gels that can be reversed by Palbocyclib. In vitro kinase assays also showed phosphorylation by CDK4. The phosphorylation sites were then identified by mutagenizing the predicted sites and phostag got to see which eliminated the shift.The authors go on to show that phosphorylation of GTSE1 affects the steady state level of the protein. Moreover, they show that expression and phosphorylation of GTSE1 confer a growth advantage on tumor cells and correlate with poor prognosis in patients.Strengths:The biochemical and mutagenesis evidence presented convincingly show that the GTSE1 protein is indeed a target of the CycD-CDK4 kinase. The follow-up experiments begin to show that the phosphorylation state of the protein affects function and has an impact on patient outcomes.Weaknesses:It is not clear at which stage in the cell cycle GTSE1 is being phosphorylated and how this is affecting the cell cycle. Considering that the protein is also phosphorylated during mitosis by CycB-Cdk1, it is unclear which phosphorylation events may be regulating the protein.

Please see point (ii) and the last paragraph in the response to Reviewer #1. Moreover, we show that, compared to the amino acids phosphorylated by cyclin D1-CDK4, cyclin B1-CDK1 phosphorylates GTSE1 on either additional residues or different sites (Figure 2H). We also show that expression of a phospho-mimicking GTSE1 mutant leads to accelerated growth and an increase in the cell proliferative index (Figure 4B,C and new Figure supplement 4D-E). Finally, we have evaluated also the cell cycle distributions by flow cytometry (new Figure supplement 4F). These analyses show that the expression of a phospho-mimicking GTSE1 mutant induces a decrease in the percentage of cells in G1 and an increase in the percentage of cells in S, similarly to what observed in AMBRA1 KO cells.

**Reviewer #3 (public review)**
Summary:This paper identifies GTSE1 as a potential substrate of cyclin D1-CDK4/6 and shows that GTSE1 correlates with cancer prognosis, probably through an effect on cell proliferation. The main problem is that the phosphorylation analysis relies on the over-expression of cyclin D1. It is unclear if the endogenous cyclin D1 is responsible for any phosphorylation of GTSE1 in vivo, and what, if anything, this moderate amount of GTSE1 phosphorylation does to drive proliferation.Strengths:There are few bonafide cyclin D1-Cdk4/6 substrates identified to be important in vivo so GTSE1 represents a potentially important finding for the field. Currently, the only cyclin D1 substrates involved in proliferation are the Rb family proteins.Weaknesses:The main weakness is that it is unclear if the endogenous cyclin D1 is responsible for phosphorylating GTSE1 in the G1 phase. For example, in Figure 2G there doesn't seem to be a higher band in the phos-tag gel in the early time points for the parental cells. This experiment could be redone with the addition of palbociclib to the parental to see if there is a reduction in GTSE1 phosphorylation and an increase in the amount in the G1 phase as predicted by the authors' model. The experiments involving palbociclib do not disentangle cell cycle effects. Adding Cdk4 inhibitors will progressively arrest more and more cells in the G1 phase and so there will be a reduction not just in Cdk4 activity but also in Cdk2 and Cdk1 activity. More experiments, like the serum starvation/release in Figure 2G, with synchronized populations of cells would be needed to disentangle the cell cycle effects of palbociclib treatment.

Please see last paragraph in the response to Reviewer #1. Concerning the experiments involving palbociclib, we limited confounding effects on the cell cycle by treating cells with palbociclib for only 4-6 hours. Under these conditions, there is simply not enough time for S and G2 cells to arrest in G1.

It is unclear if GTSE1 drives the G1/S transition. Presumably, this is part of the authors' model and should be tested.

We are not claiming that GTSE1 drives the G1/S transition (please see last paragraph in the response to Reviewer #1). GTSE1 is known to promote cell proliferation, but how it performs this task is not well understood. Our experiments indicate that, when overexpressed, cyclin D1 promotes GTSE1 phosphorylation and its consequent stabilization. In agreement with the literature, we show that higher levels of GTSE1 promote cell proliferation. To measure cell cycle distribution upon expressing various forms of GTSE1, we have now performed FACS analyses (new Figure supplement 4F). These analyses show that the expression of a phospho-mimicking GTSE1 mutant induces a decrease in the percentage of cells in G1 and an increase in the percentage of cells in S, similarly to what observed in AMBRA1 KO cells shown in the same panel and in Simoneschi et al. (Nature 2021, PMC8875297).

The proliferation assays need to be more quantitative. Figure 4B should be plotted on a log scale so that the slope can be used to infer the proliferation rate of an exponentially increasing population of cells. Figure 4c should be done with more replicates and error analysis since the effects shown in the lower right-hand panel are modest.

In Figure 4B, we plotted data in a linear scale as done in the past (Donato et al. Nature Cell Biol. 2017, PMC5376241) to better underline the changes in total cell number overtime. The experiments in Figure 4B were performed in triplicate, statistical significance was determined using unpaired T-tests with p-values<0.05, and error bars represent the mean +/- SEM. In Figure 4C, error analysis was not included for simplicity, given the complexity of the data. We have now included the other two sets of experiments (new Figure supplement 4D,E). While the effects shown in the lower right-hand panel of Figure 4C are modest, they demonstrate the same trend as those observed in the AMBRA KO cells (Figure 4C and Simoneschi et al., Nature 2021, PMC8875297). It's important to note that this effect is achieved through the stable expression of a single phospho-mimicking protein, whereas AMBRA KO cells exhibit changes in numerous cell cycle regulators. Moreover, these effects are obtained by growing cells in culture for only 5 days. A similar impact on cell growth in vivo over an extended period could pose significant risks in the long term.

**Recommendations for the authors:**

**Reviewer #1 (Recommendations for the authors):**
Figure 1E is referenced before 1D. The authors should consider switching D and E.

Done.

Figure 1D-E: The authors correctly note in the introduction that GTSE1 is encoded by a cell cycle-dependently expressed gene. Given that cell cycle genes are often associated with poor prognosis (e.g., see Whitfield et al., 2006 Nat. Rev. Cancer), this would be expected to correlate with poor prognosis. This should be mentioned in the results section.

We agree that the overexpression of certain (but not all) cell cycle-regulated genes are prognostically unfavorable across various cancer types, and we cited Whitfield et al., 2006 Nat. Rev. Cancer. However, our data indicate that phosphorylation of GTSE1 induces its stabilization and, consequently, its levels do not oscillate during the cell cycle any longer (new Figure 2G and Figure supplement 3B). Moreover, analyzing data from the Clinical Proteomic Tumor Analysis Consortium, we observed an enrichment of GTSE1 phospho-peptides (normalized to total protein) within a pan-cancer cohort as opposed to adjacent, corresponding normal tissues (Figure 2I).

Figure 2F: Contrast is too high. Blot images should not contain fully saturated black or white.

We corrected the contrast.

Figure 2G and Figure Supplement 3B: It looks like AMBRA1 KO cells do not synchronize properly in response to serum withdrawal. The cell cycle distribution should be checked by FACS. Otherwise, it is unclear whether changes in GTSE1 (phosphor) levels are only due to indirect changes in the cell cycle distribution.

Synchronization of both parental and AMBRA1 KO cells is demonstrated by the fact that the phosphorylation of Histone H3 peaks at 32 hours after serum readdition in both cases (Figure supplement 3B).

Figure 2I: It is important that phosphor-GTSE1 levels are normalized to total GTSE1 levels to understand the distinct contribution of changes in GTSE1 levels and from CCND1-CDK4 driven phosphorylation.

Done.

Figure 3A-B: These experiments should also be controlled for cell cycle distribution. Is this effect specific to GTSE1 and other AMBRA1 targets or are other G2/M cell cycle proteins also affected?

The relative short half-life of GTSE1 (<4 hours) makes its levels sensitive to acute treatments such as Palbociclib or acute AMBRA1 depletion. The effects of these treatments on GTSE1 levels are measurable within a time frame too short to significantly affect cell cycle progression. For example, we used cells with fusion of endogenous AMBRA1 to a mini-Auxin Inducible Degron (mAID) at the N-terminus. This system allows for rapid and inducible degradation of AMBRA1 upon addition of auxin, thereby minimizing compensatory cellular rewiring. Again, we observed an increase in GTSE1 levels upon acute ablation of AMBRA1 (i.e., in 8 hours) (Figure 3B), when no significant effects on cell cycle distribution are observed (please see Simoneschi et al., Nature 2021, PMC8875297 and Rona et al., Mol. Cell 2024, PMC10997477).

Figure 4: It should be noted that the correlation with cell proliferation and cell cycle protein expression is expected for any cell cycle protein, including GTSE1.

Actually, the main point of Figure 4 is to show that expression of the phospho-mimicking mutant of GTSE1 promotes cell proliferation. Comparative analysis revealed that cells overexpressing either wild-type GTSE1 or its phospho-deficient form exhibited significantly reduced proliferation rates compared to those expressing the phospho-mimicking mutant (Figure 4B,C).

The two-decades-old references 33 and 34 are not well suited to support the notion for Cyclin D1 that "the full spectrum of substrates and their impact on cellular function and oncogenesis remain poorly explored." More recent references should be used to show that this is still the case.

We added more recent references.

The authors conclude that their "data indicate that cyclin D1-CDK4 is responsible for the phosphorylation of GTSE1 on four residues (S91, S262, S454, and S724)." However, the authors' data do not exclude a role for their siblings cyclin D2, cyclin D3, and CDK6. Reflecting this, the conclusions should be toned down.

The analysis of the sites phosphorylated in GTSE1 was performed by experimentally co-expressing cyclin D1-CDK4 (Figure 2F, Figure 2H, and Figure supplement 3A), hence our statement. Yet, we agree that in cells, cyclin D2, cyclin D3, and CDK6 can contribute to GTSE1 phosphorylation.

The authors claim that they "observed that in human cells, when D-type cyclins are stabilized in the absence of AMBRA1, GTSE1 becomes phosphorylated also in G1." However, the G1-specific data presented by the authors are not controlled for, and it is unclear whether these phosphorylation events actually occur in G1 cells.

We now provide a WB in which GTSE1 phosphorylation is more evident (top panel of the new Figure 2G) (please see point (ii) in the response to the public review of Reviewer #1). This experiment clearly shows that in AMBRA1 KO cells, GTSE1 is phosphorylated at all points in the cell cycle. Synchronization of both parental and AMBRA1 KO cells is demonstrated by the fact that phosphorylation of Histone H3 peaks at 32 hours after serum re-addition in both cases (Figure supplement 3B).

**Reviewer #2 (Recommendations for the authors):**
(1) It is not clear from the presented data at which point in the cell cycle that phosphorylation of GTSE1 may be affecting the steady state level of the protein. The implication that GTSE1 is a target of CycD-CDK4 would suggest that the protein is stabilized at G1/S. Can this effect be observed?

Please see the last paragraph in the response to the public review of Reviewer #1.

(2) Considering the previous study showing that GTSE1 is also phosphorylated during mitosis by CycB-Cdk1, do levels of GTSE1 protein change during the cell cycle? Do changes in GTSE1 levels correlate with phosphorylation during the cell cycle? Cell synchronization experiments such as double thymidine and subsequent phostag analysis could shed some light on these questions.

Please see the last paragraph in the response to the public review of Reviewer #1.

(3) The authors show that the phosphomimetic mutants of GTSE1 confer a growth advantage on cells. The mechanism of this growth advantage is unclear. Is this effect due to a shorter cell cycle, enhanced survival, or another mechanism?

We did not observe increased cell survival when the phosphomimetic mutants of GTSE1 is expressed. We show that phosphorylation of GTSE1 induces its stabilization, leading to increased levels that, as expected based on the existing literature, contribute to enhanced cell proliferation. So, the role of the cyclin D1-CDK4/6 kinase-dependent phosphorylation of GTSE1 is to stabilize GTSE1.

(4) Other minor points - all of the presented immunoblots do not show molecular weight markers. The IF images require scale bars.

To prevent overcrowding of the Figures, the sizes of blotted proteins are indicated in the uncropped scans of each blot. Uncropped scans have been deposited in Mendeley at: https://data.mendeley.com/datasets/xzkw7hrwjr/1. Scale bars have been added to the IF images.